# Manic and Depressive Symptoms in Children Diagnosed with Noonan Syndrome

**DOI:** 10.3390/brainsci11020233

**Published:** 2021-02-13

**Authors:** Paolo Alfieri, Francesca Cumbo, Giulia Serra, Monia Trasolini, Camilla Frattini, Francesco Scibelli, Serena Licchelli, Flavia Cirillo, Cristina Caciolo, Maria Pia Casini, Adele D’Amico, Marco Tartaglia, Maria Cristina Digilio, Rossella Capolino, Stefano Vicari

**Affiliations:** 1Child and Adolescent Psychiatry Unit, Department of Neuroscience, Bambino Gesù Children’s Hospital, IRCCS, 00165 Rome, Italy; francesca.cumbo@opbg.net (F.C.); giulia.serra@opbg.net (G.S.); monia.trasolini@opbg.net (M.T.); cmllfrt@gmail.com (C.F.); francesco.scibelli@opbg.net (F.S.); serena.licchelli@gmail.com (S.L.); flavia.cirillo@opbg.net (F.C.); cristina.caciolo@opbg.net (C.C.); m.casini@policlinicoumberto1.it (M.P.C.); stefano.vicari@opbg.net (S.V.); 2Fondazione UILDM Lazio Onlus, 00167, Rome, Italy; 3Section of Child and Adolescent Neurology and Psychiatry, Department of Human Neuroscience, Sapienza University of Rome, 00161 Rome, Italy; 4Unit of Neuromuscular and Neurodegenerative Disorders, Department of Neuroscience, Bambino Gesù Children’s Hospital, IRCCS, 00165 Rome, Italy; adele2damico@opbg.net; 5Genetics and Rare Diseases Research Division, Bambino Gesù Children’s Hospital, IRCCS, 00146 Rome, Italy; marco.tartaglia@opbg.net (M.T.); mcristina.digilio@opbg.net (M.C.D.); rossella.capolino@opbg.net (R.C.); 6Department of Life Sciences and Public Health, Università Cattolica del Sacro Cuore, 00168 Rome, Italy

**Keywords:** psychopathological features, attention deficit and hyperactivity disorder, deficient emotional self-regulation, irritability, anxiety traits, mood

## Abstract

Noonan syndrome (NS) is a dominant clinically variable and genetically heterogeneous developmental disorder caused by germ-line mutations encoding components of the Ras–MAPK signaling pathway. A few studies have investigated psychopathological features occurring in individuals with NS, although they were poorly analyzed. The aim of the present work is to investigate the psychopathological features in children and adolescents with NS focusing on depressive and hypo-manic symptoms. Thirty-seven subjects with molecularly confirmed diagnosis were systematically evaluated through a psychopathological assessment. In addition, an evaluation of the cognitive level was performed. Our analyses showed a high recurrence of attention deficit and hyperactivity disorder symptoms, emotional dysregulation, irritability, and anxiety symptomatology. The mean cognitive level was on the average. The present study provides new relevant information on psychopathological features in individuals with NS. The implications for clinicians are discussed including the monitoring of mood disorders in a clinical evolution.

## 1. Introduction

Noonan syndrome (NS) is a clinically variable and genetically heterogeneous disorder affecting development and growth. Major features include a distinctive facial appearance, short stature, congenital cardiac defects, pulmonary valve stenosis most commonly, and hypertrophic cardiomyopathy (HCM, 10–20%) as well as variable cognitive deficits [1,2,3,4]. NS is considered as one of the most common non-chromosomal disorders affecting neurodevelopment.

NS is caused by germ-line mutations in genes encoding components of the Ras–MAPK signaling pathway [4]. In approximately 50% of cases, it is caused by dominant mutations in *PTPN11* [5]. Heterozygous activating mutations in *SOS1*, *RAF1*, and *RIT1* also represent relatively common events, while an increasing number of genes (e.g., *NRAS*, *SOS2*, *RRAS2*, and *MRAS*) have been reported to account for a relatively small proportion of cases [6,7,8,9]. In a significant proportion of patients with NS, loss-of-function (recessive form) or dominant negative (dominant form) mutations in *LZTR1* have been reported. NS is clinically related to other developmental disorders collectively denominated “RASopathies”, sharing the upregulation of the Ras–MAPK signaling cascade as a mechanism of disease [10,11,12]. Among these, Costello syndrome (CS), cardiofaciocutaneous syndrome (CFCS), Noonan syndrome with multiple lentigines (NSML, formerly known as LEOPARD syndrome), Mazzanti syndrome (MS), neurofibromatosis type 1, and Legius syndrome have been characterized [13]. New conditions are also emerging [10]. In RASopathies, a cognitive deficit significantly varies depending on the gene involved and the type of mutation. Most patients with NS show an intellectual cognitive profile in the low/average range [14,15,16]. While some studies described a relatively low occurrence of intellectual disability (11% of cases) in *SOS1* patients with respect to the NS general population [17,18], other significant genotype–phenotype correlations and relatively other neuropsychiatric features are less clear, probably due to the relatively small size of the cohorts analyzed [19]. Psychopathological features of individuals with NS have also been investigated, and a relatively higher risk of psychiatric diseases when compared to the control population have been reported [20,21,22,23]. Mood disorders, social problems, communication difficulties, executive functioning impairment, and attention deficit have been listed as major features [20,24,25].

During adulthood, individuals diagnosed with NS struggle with well-known socialization and daily living problems, including difficulties with maintaining employment and typical social relationships [1,21]. Some cases of psychiatric disorders (mostly in subjects with mild cognitive deficits), such as bipolar affective disorder [21,26], schizophrenia [21,27], panic disorder, and alexithymia [21,28], have been reported.

Recent works [29] confirmed emotional and internalizing dysfunctions using a self-report questionnaire in adults with NS. Some studies highlighted high levels of demoralization in adults with NS, presumably due to the burden of chronic disorder. Furthermore, authors found low levels of life satisfaction (including a high percentage of bullying episodes in childhood). Broadly speaking, individuals with NS showed an introvert personality style combined with internalizing difficulties. Moreover, other studies reported a that a high percentage of depression and anxiety disorders is estimated in 49% of subjects diagnosed with NS [4,30,31].

Less is known about affective disorders in NS in childhood. It is well established that children and adolescences with NS mostly suffer from attention deficit and hyperactivity (ADHD) symptoms (up to 40–50%) [19,25,32]. Individuals with ADHD generally appear as highly impulsive with difficulties in self-regulatory functions and in emotional regulation [33,34]. The emotional regulation is the ability to “(a) inhibit inappropriate behavior related to strong negative or positive emotions, (b) self-soothe any physiological arousal that the strong affect has induced, (c) refocus attention, and (d) organize for coordinated action in the service of an external goal [35,36]”. Furthermore, patients with ADHD often show a range of affective symptoms such as emotional instability, impulsivity, agitation, restlessness, and mood dysregulation [37]. In order to plan an appropriate treatment and inform patients and their parents about the psychopathological development and long term prognosis affective symptoms during childhood in NS, characterizing mood dysregulation in juvenile subjects as a typical of ADHD or as a separate psychiatric condition would be very interesting [36,38].

Several studies [39,40,41,42] analyzed emotional dysregulation using the Child Behavior Checklist (CBCL)-AAA profile [36]. This score, calculated as the sum of Anxious/Depressed, Attention, and Aggression, is a widely used assessment for the Deficient Emotional Self-Regulation (DESR) evaluation which has been frequently identified in ADHD subjects showing severe forms of mood and behavioral dysregulation [37,39].

Our previous findings [43] also showed affective and behavioral dysregulation symptoms as measured using the CBCL-AAA profile in children diagnosed with RASopathies (mainly NS). Several studies underlined a strong overlap between the ADHD and anxious-depressive features among children diagnosed with RASopathies, mainly among NS subjects [44,45,46]. Despite that, little is known about the presence of mood and anxiety symptoms in individuals with NS.

The aim of the present study is to investigate the prevalence and severity of depressive, dysphoric/mixed, and (hypo)manic features in a cognitively assessed sample of 37 subjects with NS using a neuropsychiatric evaluation, parent-report questionnaire, and semi-structured interviews according to the Diagnostic and Statistical Manual of Mental Disorders, fourth edition, text revision (DSM-IV-TR) criteria [47].

## 2. Materials and Methods

### 2.1. Participants

The cohort was recruited from the Child and Adolescent Psychiatry Unit of the Bambino Gesù Children Hospital, Rome, from 2017 to 2020. A structured assessment was performed during a 3-day evaluation process. The Best Estimate Consensus Diagnosis based on DSM-IV-TR criteria [47] were performed by multidisciplinary groups composed at least by one child psychiatrist and one clinical licensed psychologist according to parent report questionnaires, semi-structured interviews, clinical history, and medical records. Parents or the legal representative of the subjects provided a written, informed consent at a clinic intake for potential research analysis and anonymous reporting of findings in an aggregate form, in accordance with Italian legal and ethical requirements for clinical data.

Thirty-seven individuals with NS were assessed by a multi-method procedure that includes self-report tools and semi structured interviews. Both subjects and at least one of the parents completed the psychopathological assessment. A diagnosis of NS was made by experienced medical geneticists and pediatricians based on a clinical evaluation and was confirmed by molecular analysis.

### 2.2. Structured Assessment

General cognitive abilities were assessed by means of appropriate developmental cognitive tools including Wechsler Intelligence Scale for Children, fourth edition [48], Leiter International Performance Scale Revised (Leiter-R) VR Battery [49], and Raven Coloured Progressive Matrices Test [50]. We classified Intellectual abilities according to DSM-IV-TR [47].

The Schedule for Affective Disorders and Schizophrenia for School Age Children, present and lifetime version (K-SADS-PL) [51] is a semi-structured interview used to assess current and past psychopathological features and psychiatric disorders in children and adolescents according to the DSM-IV-TR criteria [47].

The Children’s Depression Rating Scale—Revisited (CDRS-R) [52] is a semi-structured interview used to rate depressive symptoms for ages 6–18 years on 16 items (rated 0 to 3, 4, 5 or 6) with a total score of 18–120 (cut-off clinical positivity > 55). The assessment explores 17 scales: School dysfunctions; difficulties in having fun; difficulties in interpersonal relationships; sleep disorders; appetite disorders; excessive fatigue; psychosomatic complaints; irritability; excessive guilt; low self-esteem; depressive feelings; morbid ideas; suicidal ideation; excessive crying; reduced facial expressions; slow speech; and motor hypoactivity. The CDRS-R clinical positivity cut-offs for the subscale were: <3 normal scores; from 3 to 4 borderline scores; and >4 clinical scores. “Sleep Disturbance”, “Appetite Disturbance”, and “Listless Speech” scales have different cut-offs: <3 normal scores; 3 borderline scores; and >3 clinical scores.

The Kiddie—SADS—Mania Rating Scale (K-SADS-MRS) [53] is a structured interview rating scale used to rate manic symptoms in children and adolescents, composed by 14 scales: Euphoria and expansiveness; irritability and anger; mood liability, reduced need for sleep; crowding thoughts; increased energies; increased activities; motor hyperactivity; grandiosity; speech; distractibility; poor judgment; hallucinations; delusions (considered positive if >12). The K-SADS-MRS clinical positivity cut-offs were: >12 clinical scores; subscales cut-off <2 normal scores; subscales cut-off 2/3 for borderline scores; and subscales cut-off >3 clinical scores.

The Italian version of the Child Behavior Checklist for ages 6 to 18 years (CBCL/6–18) [54] was completed by the subject’s caregivers to rate behavioral and emotional problems in children and adolescents. The CBCL 6-18 is an extensively used tool that provides scores for three broadband behavior rating scales: Internalizing symptoms, externalizing symptoms, and total behavioral problems. The sub-items of these three broadband scales included the eight syndrome scales (withdrawn/depressed, somatic complaints, anxious/depression, social problems, thought problems, attention problems, rule-breaking behaviour, and aggressive behaviour). The CBCL AAA profile was calculated by examining ratings on attention problems, aggression and anxious/depressed scales, is defined as deficient emotional self-regulation (DESR) by a score of ≥180 and <210 (1 < SD < 2) and as a dysregulation profile (DP) by a score of >210 (>2 SD) on the sum of three syndromic scales [39].

Participants were globally evaluated by an expert child psychiatrist investigating psychopathological signs/psychiatric disorders according to the criteria of DSM-IV-TR [47], and assessed with rating scales by experienced psychologists. Participants with “traits of disorder” refer to a subsyndromal symptomatology (i.e., NOT fulfilling the DSM-IV-TR criteria for ADHD or anxiety disorders).

## 3. Results

### 3.1. Sample Characteristics

The study included 37 subjects with a genetically confirmed diagnosis of NS. Subjects had a mean (M) age of 11.7 and a standard deviation (SD) ± 2.8 (age range 7–18 years), 24 (65%) were males. Twenty-four subjects were heterozygous for mutations in *PTPN11*, seven carried mutations in *SOS1*, two in *LZTR1*, two in *RAF1*, and one subject was heterozygous for a mutation in *RIT1*. One subject carried a *SHOC2* mutation, which underlies the Mazzanti syndrome, a phenotype clinically related to NS. The cognitive level ranged from moderate impairment to cognitive level on the average (M ± SD: 88.3 ± 13.4; minimum: 48; maximum; 110) (Table 1).

The Best Estimate Consensus Diagnosis has revealed that 21 (60%) subjects were diagnosed with ADHD and two subjects showed ADHD traits (6%), 10 (27%) subjects were diagnosed with anxiety disorder or anxiety traits, three (9%) with dysthymia, and one subject was diagnosed with a separation anxiety disorder. Only one patient did not have any psychopathological disorder. A synopsis of prevalence disorders is available in Table 1.

### 3.2. Manic and Depressive Symptoms across Groups

Descriptive statistics were calculated for CDRS-R total scores, K-SADS-MRS scores, and AAA profile scores. To better characterize the presence of depressive and manic symptoms across clinical groups, the percentage of participants above and below the cut-off of CDRS-R (CDRS-R+ and CDRS-R−, respectively) and K-SADS-MRS (K-SADS-MRS+ and K-SADS-MRS−, respectively) were elaborated. About half of the subjects (*n* = 18, 49%) had clinically significant depressive symptoms as rated with the CDRS-R scale, and the CDRS-R total score was M ± SD: 53.6 ± 8.4.

Sixteen percent of the subjects (*n* = 6, 16%) had clinically significant (hypo)manic symptoms as rated with the K-SADS-MRS scale, and the K-SADS-MRS total score was M ± SD: 6.73 ± 4.30.

Moreover, almost half of the subjects (*n* = 16, 46%) showed a clinically significant Emotional Dysregulation as measured with the CBCL-AAA profile, with a CBCL-AAA score of M ± SD: 184.4 ± 21.9 (see Table 2).

Figure 1 and Figure 2 show a prevalence of depressive symptoms, in CDRS-R+ and CDRS-R− groups, respectively, while Figure 3 and Figure 4 show a prevalence of manic symptoms in K-SADS-MRS+ and K-SADS-MRS− groups, respectively.

#### 3.2.1. Depressive Symptoms

As stated, above eighteen individuals (49%) were considered to have significant depressive symptoms as rated with the CDRS-R rating scale (CDRS–R score >55; total score M ± SD: 60 ± 3.9). The subscales suggestive of clinical depressive symptomatology were: “Impaired schoolwork”, “Difficult having fun”, “Irritability”, “Low self-esteem”, and “Depressive feeling”. Specifically, one individual (5.6%) reported clinical scores, and eight (44.4%) reported borderline scores in the “Impaired schoolwork” subscale (M ± SD: 2.3 ± 1.4); four individuals (22.2%) reported clinical scores, and six individuals (33.3%) reported borderline scores in the “Difficult having fun” subscale (M ± SD: 3.0 ± 1.0); four individuals (22.2%) reported clinical scores and 11 patients (61.1 %) reported borderline scores in the “Irritability” subscale (M ± SD: 3.27 ± 1.5); three patients (16.7%) reported clinical scores and six (33.3%) obtained borderline scores in the “Low self-esteem” subscale (M ± SD: 2.7 ± 1.4); 11 patients (61.1%) obtained borderline scores in the “Depressive feeling” subscale (M ± SD: 2.6 ± 0.10). Among all those who presented a clinical depressive symptomatology, four individuals (22.2%) have reported a clinically significant suicidal ideation (M ± SD: 1.5 ± 0.8) (see Figure 1).

Nineteen individuals (51%) obtained a normal score (CDRS-R− < 55; a total score of M ± SD: 47.5 ± 6.9). Despite this, the subscales “Irritability” and “Low self-esteem” were suggestive of clinical depressive symptomatology. In particular, one individual (5.3%) obtained a clinical score and 12 individuals (63.2%) obtained borderline scores in the “Irritability” subscale (M ± SD: 2.9 ± 0.10); two individuals (10.5%) obtained clinical scores and seven (36.8%) obtained borderline scores in the “Low self-esteem” subscale (M ± SD: 2.4 ± 1.3) (see Figure 2).

#### 3.2.2. Manic Symptoms

As stated above, six individuals (16%) obtained clinically significant scores (K-SADS-MRS+ > 12 total score M ± SD: 15.16 ± 2.13). The subscales suggestive of clinical (hypo) manic symptoms were: “Euphoria expansiveness”, “Irritability and anger”, “Mood liability”, “Unusually energetic”, “Increase in activities”, “Motor hyperactivity”, “Speech”, and “Distractibility”. Specifically, one individual (16.7%) reported a clinical score, and three (50%) obtained a clinical score in the “Euphoria expansiveness” subscale (M ± SD: 2.16 ± 1.16); two individuals (33.3%) obtained clinical scores and four individuals (66.7%) obtained borderline scores in the “Irritability and anger” subscale (M ± SD: 3.3 ± 0.5); two individuals (33.3%) obtained clinical scores and four individuals (66.7%) obtained borderline scores in the “Mood liability” subscale (M ± SD: 3.3 ± 1.03); six individuals (100%) obtained borderline scores in the “Unusually energetic” subscale (M ± SD: 2.3 ± 0.5); four individuals (66.7%) obtained borderline scores in the “Increase in activities” subscale (M ± SD: 1.7 ± 0.5); two individuals (33.3%) obtained clinical scores and four individuals (66.7%) obtained borderline scores in the “Motor hyperactivity” subscale (M ± SD: 3.3 ± 0.5); four individuals (66.7%) obtained borderline scores in the “Speech” subscale (M ± SD: 2 ± 0.8) and five (83.3%) obtained borderline scores in the “Distractibility” subscale (M ± SD: 2 ± 0.6) (see Figure 3).

Thirty-one individuals (84%) obtained a normal score (K-SADS-MRS+ < 12). Despite that, the subscales “Irritability and anger” and “Mood liability” were suggestive of clinical (hypo) manic symptoms. Specifically, two individuals (6.4%) obtained clinical scores and 27 (87.1%) obtained borderline scores in the “Irritability and anger” subscale (M ± SD: 2.5 ± 0.8); one (3.2%) obtained a clinical score and 17 (54.8%) obtained borderline scores in the “Mood liability” subscale (M ± SD: 2 ± 0.9) (see Figure 4).

It is worth noting that the scores above the borderline/clinical cut-off were found in the “Irritability” CDRS-R scale and “Irritability and Anger” K-SADS-MRS, respectively in 75.67% and 94.59% of the whole sample.

## 4. Discussion

The present study aimed to provide a detailed description of psychopathological features in children and adolescents with NS. To the best of our knowledge, this is the first study reporting data on the development of mood symptoms and emotional dysregulation in children diagnosed with NS using a “gold standard” test battery for mood disorder rating.

Consistently with previous studies that were demonstrated in patients with NS, the high frequency of hyperactivity and impairment in attention skills and executive functioning [3,15,20,21,24,55,56], 66% of the subjects were diagnosed with ADHD diagnosis or traits.

Although only 9% of the subjects were diagnosed with a mood disorder (dysthymia) following the DSM-IV criteria, half of the children (*n* = 18, 49% of the total sample) were found to have a clinically significant depressive symptomatology when rated with the CDRS-R, the “gold standard” rating scale to assess depressive symptoms in children and adolescents (Table 2, Figure 1). Among subjects with a clinically meaningful depressive symptomatology, the irritability, anhedonia, school impairment, low self-esteem, and depressive feelings were the most commonly reported symptoms (Table 2, Figure 1). Furthermore, about 22% of subjects with a clinical depressive symptomatology reported suicidal ideation (see Figure 1). Of note, irritability and low self-esteem were rated as clinically significant in 73% and 50%, respectively of the NS subjects in our sample.

We identified that 16% of the subjects showed clinically significant scores at the manic symptoms’ rating with the K-SADS-MRS. The more frequently reported symptoms were irritability, mood lability and euphoria, increased energy and activity, motor hyperactivity and pressured speech, as well as mood lability. Among these, about two thirds of the subjects were found to be frequently euphoric, almost all of them scored significantly in irritability and mood lability, and borderline in highly energetic. Of note, 95% of the total number of subjects reported a clinically significant irritability and 65% a clinically significant mood lability, as rated with the K-SADS-MRS.

Those findings underline an impressive prevalence of subsyndromal manic, depressive, and mixed symptomatology in children with NS, mostly clinically diagnosed with ADHD, but showing mood and behavioral alterations, with an uncommon similarity among the subjects, and far beyond the simple clinical picture of a classic ADHD.

Indeed, nearly a half (46%) of our cohort with NS shows at least a DESR profile (dysregulation profile or deficient emotional self-regulation), highlighting emotional dysregulation problems as a relatively common feature. This latter finding confirms the general clinical evidence that subjects with ADHD (with and without genetic syndromes) often show emotional dysregulation [37,39,43,44,45,46], and raise questions on the psychopathological characterization and the future development of this unspecific clinical feature. Some authors have reported that a very severe dysregulation profile rated with CBCL-AAA is highly related with the progression to a full syndromic pediatric bipolar disorder, frequently associated with ADHD [57]. The unusual high prevalence of mood symptoms such as anhedonia, low self-esteem, mood lability, and irritability in this sample of NS subjects, together with ADHD and severe emotional dysregulation need further evaluation to assess if they can be considered phenotypical characteristics of this genetic syndrome [23].

Our findings also highlighted the presence of depressive symptoms (nearly 50% with scores above the cut-off in CDRS-R) and manic symptoms (around 16%), even if only 9% of the individuals in our sample satisfied the criteria for mood disorders (dysthymia) according to DSM-IV-TR. The lower percentage of mood disorders clinically diagnosed in our cohort could be due to the relatively young mean age (11.7 years) of our subjects and to the probable tendency towards an underdiagnosis of mood disorders when they onset with a prevalent mood dysregulation, unclear episodes, and rapid mood shifts. The high prevalence of irritability, mood lability with rapid mood shifts, and subsyndromal depressive symptoms such as anhedonia and low self-esteem (see Figure 1, Figure 2, Figure 3 and Figure 4) have been largely reported as a risk factor for the development of major depressive and bipolar disorders later in life, frequently associated with high morbidity, disability, and comorbid substance and anxiety disorders [58]. Given these associations [23], a careful psychiatric follow-up of these symptoms is important over time.

Another important finding is that 28% of our sample have anxious symptomatology, consistent with previous findings describing a high prevalence of anxiety traits and ADHD in children with NS when compared to the general population [19]. Several authors have reported that anxious symptoms are important predictors of future development of depressive disorders, and an increased risk of worse outcomes of depressive episodes in terms of greater depressive long term morbidity as well as an increased risk in suicidal behavior [44,45,46,59]. Indeed, several studies have reported a high prevalence of depressive disorders in adults diagnosed with NS [1,21,29], and a high prevalence of NS subjects have been reported to need an antidepressant treatment during adult ages [21].

Considering the large amount of evidence of a high risk for depressive episodes in adults with NS, we hypothesized that there might be a continuity between the severe mood-dysregulation with subsyndromal depressive and (hypo)manic symptoms observed in childhood [15,18,19,20,24,25,32,36,44], and the depressive episodes described in adults with NS. Longitudinal studies are needed to explore whether irritability and the DESR profile contribute to mood disorders in children with NS, and whether they have a predictive value of future development of a mood disorder in adult ages.

## 5. Conclusions

In conclusion, this research highlights the presence of subsyndromal depressive symptoms (irritability, anhedonia, school impairment, low self-esteem, depressive feelings, suicidal ideation), as well as (hypo)manic symptoms (irritability, mood lability, euphoria, increased energy and activity, motor hyperactivity, pressured speech, and mood lability) in NS. Furthermore, nearly a half of our patients showed a dysregulation profile or deficient emotional self-regulation.

Despite the fact that ADHD is the most frequent diagnosis in our sample, the presence of subsyndromal depressive and (hypo)manic symptoms seems to be far beyond the simple clinical picture of a classic ADHD. A longitudinal observation of clinical symptoms’ evolution in these patients would allow a further comprehension of the linkage between the combination of symptoms and clear depressive or manic diagnoses.

## 6. Limitations

Our study is not without limitations, the major being an absence of a control sample. Furthermore, the relatively small size of our sample does not allow for a systematic statistical analysis of the correlation between the genotype and behavioural phenotype. Moreover, given the relatively small sample size, gender differences have not been explored in this study. Finally, “hyperactivity” has been addressed by means of a semi-structured interview (K-SADS-PL) [51]. The hyperactivity diagnosis has been performed by multidisciplinary groups composed at least of one child psychiatrist and one clinical licensed psychologist according to the DSM-IV-TR criteria [47] based on semi-structured interviews and clinical history. However, further studies using specific diagnostic tools (such as Conners Parent Rating Scale-Revised and Conners Teacher Rating Scale-Revised) [60] would improve the characterization of these symptoms.

## Figures and Tables

**Figure 1 brainsci-11-00233-f001:**
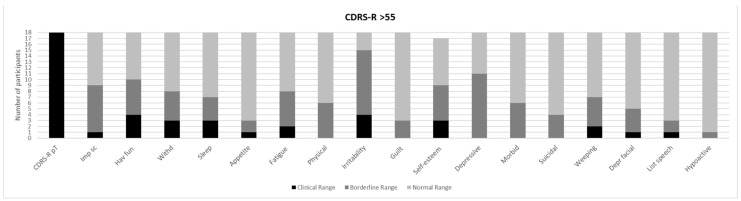
Prevalence of depressive symptoms in the children’s depression rating scale (CDRS-R) > 55 (CDRS-R+). Imp sc: Impaired schoolwork; Hav fun; Difficult having fun; Withd: Social withdrawal; Sleep: Sleep disturbance; Appetite: Appetite disturbance; Fatigue: Excessive fatigue; Physical: Physical complaints; Guilt: Excessive guilt; Self-esteem: Low self-esteem; Depressive: Depressive feeling; Morbid: Morbid ideation; Suicidal: Suicidal ideation; Weeping: Excessive weeping; Depr facial: Depressed facial affect; List speech: Listless speech.

**Figure 2 brainsci-11-00233-f002:**
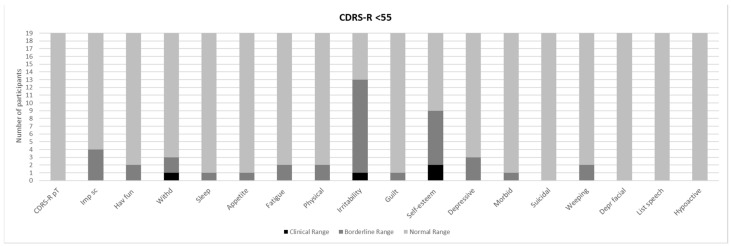
Prevalence of depressive symptoms in CDRS-R < 55 (CDRS-R-). Imp sc: Impaired schoolwork; Hav fun; Difficult having fun; Withd: Social withdrawal; Sleep: Sleep disturbance; Appetite: Appetite disturbance; Fatigue: Excessive fatigue; Physical: Physical complaints; Guilt: Excessive guilt; Self-esteem: Low self-esteem; Depressive: Depressive feeling; Morbid: Morbid ideation; Suicidal: Suicidal ideation; Weeping: Excessive weeping; Depr facial: Depressed facial affect; List speech: Listless speech.

**Figure 3 brainsci-11-00233-f003:**
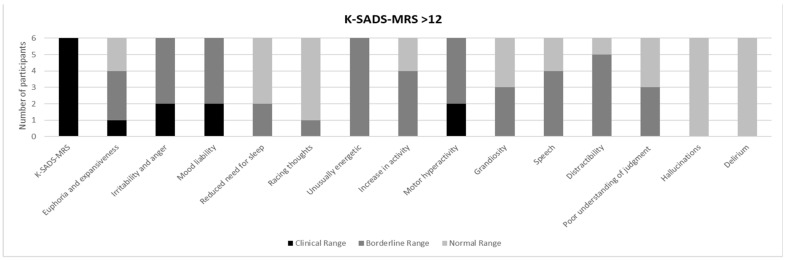
Prevalence of manic symptoms in the schedule for affective disorders and schizophrenia for school age children (K-SADS-MRS) > 12 (K-SADS-MRS+).

**Figure 4 brainsci-11-00233-f004:**
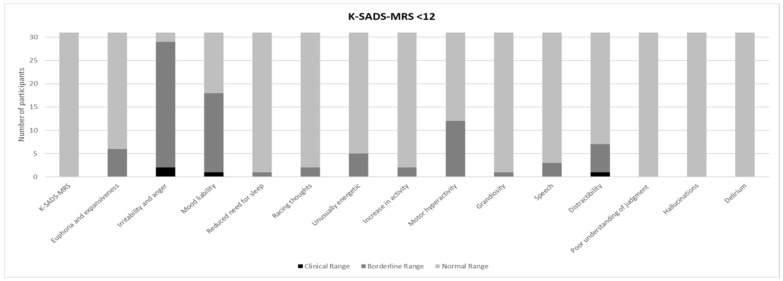
Prevalence of manic symptoms in K-SADS-MRS < 12 (K-SADS-MRS-).

**Table 1 brainsci-11-00233-t001:** Descriptive characteristics of subjects with Noonan syndrome.

**Age**, mean (sd)		11.7 (2.8)
**IQ**, mean (sd)		88.3 (13.4)
**Sex**, N (%)	Male	24 (64.9)
	Female	13 (35.1)
**Mutation**, N (%)	*LZTR1*	2 (5.4)
	*PTPN11*	24 (64.9)
	*RAF1*	2 (5.4)
	*RIT1*	1 (2.7)
	*SCHOC2*	1 (2.7)
	*SOS1*	7 (18.9)
**Diagnosis**, N (%)	No-one	1 (2.9)
	Anxiety traits	5 (13.5)
	ADHD traits	2 (5.7)
	ADHD	21 (60)
	Generalized Anxiety Disorder	4 (11.4)
	Dysthymia	3 (8.6)
	Separation anxiety	1 (2.9)

Legend: N: Number; sd: Standard deviation; IQ: Intelligence quotient.

**Table 2 brainsci-11-00233-t002:** Descriptive statistics for the questionnaire and interviews score.

Depression Severity, N (%)	
*CDRS-R+* (*T-score* ≥ 55)	18 (48.6)
*CDRS-R−* (*T-score* < 55)	19 (51.4)
Depression Severity, mean (sd)	
*CDRS-R total score*	53.6 (8.4)
Mania/Hypomania Symptoms, N (%)	
*K-SADS-MRS+* (≥ 12)	6 (16.2)
*K-SADS-MRS−* (<12)	31 (83.8)
Mania/Hypomania Symptoms, mean (sd)	
*K-SADS-MRS total score*	6.73 (4.3)
AAA Profile, N (%)	
*Deficient Emotional Self-Regulation* (*DERS*; ≥180 < 210)	11 (30)
*Dysregulation Profile* (*DP*; > 210)	6 (16)
AAA Profile, means (sd)	
*AAA score*	184.4 (21.9)

Legend: N: Number; sd: Standard deviation; CDRS-R: Children’s Depression Rating Scale—Revisited; K SADS-MRS: Kiddie—SADS—Mania Rating Scale; AAA: Anxious/Depressed, Attention, and Aggression.

## Data Availability

The data that support the findings of this study are available on request from the corresponding author. The data are not publicly available due to privacy or ethical restrictions.

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
