# Peer review of "Manic and Depressive Symptoms in Children Diagnosed with Noonan Syndrome"

_brainsci, 2021, doi:10.3390/brainsci11020233_

Round 1

Reviewer 1 Report

Congratulations on your study reporting data on the development of mood symptoms and emotional dysregulation in children diagnosed with NS.

Congratulations on your study reporting data on the development of mood symptoms and emotional dysregulation in children diagnosed with NS. Though there was no control group,  standardized rating scales were used. Though the sample size was limited, getting data on the subset of Noonan's syndrome patients is not an easy task. 

Please revise

1) Table 1 so that it is more clear. Sex, mutation, and diagnosis need to be segregated from other variables for clarity.

2) Table 2 so that Depression Severity, N (%) does not appear as a table header

Please expand the limitations of your study and review your draft for grammatical improvements.

Author Response

Congratulations on your study reporting data on the development of mood symptoms and emotional dysregulation in children diagnosed with NS.

Congratulations on your study reporting data on the development of mood symptoms and emotional dysregulation in children diagnosed with NS. Though there was no control group,  standardized rating scales were used. Though the sample size was limited, getting data on the subset of Noonan's syndrome patients is not an easy task.

Author answer: We wish to thank the reviewer for her/his positive assessment despite study’s limitations. We are happy to provide an uploaded version of the paper based on those remarks.

Please revise

1) Table 1 so that it is more clear. Sex, mutation, and diagnosis need to be segregated from other variables for clarity.

Author answer: we modified the table accordingly.

2) Table 2 so that Depression Severity, N (%) does not appear as a table header

Author answer: we have modify the table following the reviewer’s comments

3) Please expand the limitations of your study and review your draft for grammatical improvements

Author answer: we have expanded limitations of our study as follow:

“Our study is not without limitations, with the major being the absence of a control sample. Furthermore, the relatively small size of our sample does not allow for systematic statistical analysis of the correlation between the genotype and behavioural phenotype. Moreover, given the relatively small sample size gender differences have not been explored in this study. Finally, “hyperactivity” have been addressed by means of semi-structured interview (K-SADS-PL). Hyperactivity diagnosis has been performed by multidisciplinary groups composed at least by one child psychiatrist and one clinical licensed psychologist according to semi-structured interviews and clinical history. However, further studies using specific diagnostic tools (such as Conners Parent Rating Scale-Revised, Conners Teacher Rating Scale-Revised) would improve this characterization“

Following the remark, the manuscript has been revised for catching typos and grammatical errors.

Reviewer 2 Report

The manuscript by Alfieri et al., was a relatively comprehensive research article on the potential roles of psychiatric involvement in the children populations. The authors focused on Noonan Syndrome and related genotypes and tested the depressive behaviors. Experiments were well performed and data was well collected.  There were some moderated concerns:

  • It was noticed that introductions were included for risk genes (Lines 41-46), the specificity of genotypes over other neuropsychiatric disorders should be described in details.
  • The gender/sex differences not addressed but important for studying neuropsychiatry.
  • Lines 256-258, “hyperactivity”was mentioned but it was not measured in the study. Based on the data, new analysis should be performed.

Author Response

The manuscript by Alfieri et al., was a relatively comprehensive research article on the potential roles of psychiatric involvement in the children populations. The authors focused on Noonan Syndrome and related genotypes and tested the depressive behaviors. Experiments were well performed and data was well collected. 

Author’s answer: We thank the reviewer for the positive feedback and the constructive comments. We are happy to provide an uploaded version of the paper based on your observations

There were some moderated concerns:

  • It was noticed that introductions were included for risk genes (Lines 41-46), the specificity of genotypes over other neuropsychiatric disorders should be described in details.

Author’s answer: In this revised version of the manuscript, a sentence (lines 54-58) introducing genotype-phenotype correlation studies for cognitive and neuropsychiatric features has been included in the “Introduction” section.

  • The gender/sex differences not addressed but important for studying neuropsychiatry.

Author’s answer: The gender/sex differences have not been described in the paper given that analysis performed did not show significant gender differences regarding affective symptoms

  • Lines 256-258, “hyperactivity”was mentioned but it was not measured in the study. Based on the data, new analysis should be performed.

Author’s answer: “Hyperactivity” have been addressed by means of semi-structured interview (K-SADS-PL). ADHD diagnosis, present in 60% of our sample (see table 1), has been performed by multidisciplinary groups composed at least by one child psychiatrist and one clinical licensed psychologist according to DSM-IV-TR criteria based on semi-structured interviews and clinical history. However, further studies using specific diagnostic tools (such as Conners Parent Rating Scale-Revised, Conners Teacher Rating Scale-Revised) would improve characterization of these symptoms.

Round 2

Reviewer 2 Report

The manuscript by Alfieri et al., was a relatively comprehensive research on the potential effects of mental illnesses in the children patients with Noonan syndrome. The authors focused on the genetic variations and studied the psychiatric symptoms. There were some minor concerns:

  • Line 324 rationale of being “strictly psychiatrically monitored” was unknown.
  • Justification of potential differences or the scientific message needed regarding ADHD-like symptoms.
  • Title needs to be changed since hyperactivity has also been measured.

Author Response

The manuscript by Alfieri et al., was a relatively comprehensive research on the potential effects of mental illnesses in the children patients with Noonan syndrome. The authors focused on the genetic variations and studied the psychiatric symptoms. There were some minor concerns:

  • Line 324 rationale of being “strictly psychiatrically monitored” was unknown.

Author’s answer: We have amended this sentence as follow:

Given these associations [23], it is important a careful psychiatric follow-up of these symptoms over the time”

  • Justification of potential differences or the scientific message needed regarding ADHD-like symptoms.

Author’s answer: ADHD-like symptoms refers to participants with subsyndromal symptomatology (i.e. NOT fulfilling DSM-IV-TR criteria for ADHD disorder). Explanation of this concept is present in Materials and Methods section.

  • Title needs to be changed since hyperactivity has also been measured:

Author’s answer: ADHD was evaluated clinically and by structured assessment using K-Sads-PL. Specific gold standard measures have been used to assess Manic and Depressive Symptoms. The title reflect these two main manifestations.